# Using Relational Community Engagement within the Digital Health Intervention (DHI) to Improve Access and Retention among People Living with HIV (PLWH): Findings from a Mixed-Method Study in Cambodia

**DOI:** 10.3390/ijerph20075247

**Published:** 2023-03-23

**Authors:** Kennarey Seang, Sovathana Ky, Bora Ngauv, Sovatha Mam, Vichea Ouk, Vonthanak Saphonn

**Affiliations:** 1Grant Management Office, University of Health Sciences, Phnom Penh 12201, Cambodia; 2National Center for HIV/AIDS, Dermatology and STDs, Phnom Penh 121002, Cambodia; 3Rectorate, University of Health Sciences, Phnom Penh 12201, Cambodia

**Keywords:** COVID-19, HIV/AIDS, complexity-based method, digital health intervention, DHI, eHealth

## Abstract

We examined the impact of COVID-19-associated restrictive measures on the HIV care system in Cambodia through a complexity lens and aimed to use the findings to integrate social and relational processes into the design and implementation of proposed solutions that could support program outcomes during these times. Through a mixed-method design, we generated data on the strength of connection and quality of relationships between stakeholders and how this, in turn, provided a more holistic understanding of the challenges experienced during a pandemic. We interviewed 43 HIV care providers and 13 patients from eight HIV clinics and 13 policy-level stakeholders from relevant institutions involved in HIV care from April to May 2021. We identified several challenges, as well as an opportunity to improve HIV care access that built upon a strong foundation of trust between the HIV care providers and receivers in Cambodia. Trusting relationships between providers and patients provided the basis for intervention development aiming to improve the care experience and patients’ engagement in care. Iterative research processes could better inform the intervention, and communication resources provided through relational skills training are key to their application and sustainability.

## 1. Introduction

Electronic health services have transformed the ways in which patients are engaged in care and, according to the World Health Organization (WHO), Electronic Health (eHealth) plays an important role in providing affordable care to underserved populations [1]. Addressing the gaps in Universal Health Coverage (UHC) requires the evolution of care delivery models and solutions to shortages in the health workforce, both of which could potentially be achieved through digital health solutions [2]. As a result, in recent years there had been evidence of the increased use of digital health services across the globe and of their benefits in optimizing STIs/HIV prevention and treatment strategies [1,3]. 

At the end of 2020, the WHO Western Pacific Regional Office (WPRO) commissioned a multi-country community engagement (CE) research initiative, aiming to highlight the importance of complexity-informed research processes in which relational dimensions between various stakeholders within the health system were examined and used to inform and improve this community participatory approach of planning, designing and implementing the intervention [4]. In Cambodia, the research team from the University of Health Sciences (UHS) led the CE work in 2020–2021, as well as the extended activities of this work in 2022, in order to improve access and retention among people living with HIV (PLWH). As the world population began reporting negative effects of the coronavirus diseases 2019 (COVID-19) pandemic [5,6], our proposed CE work initially aimed at understanding the barriers to comprehensive HIV care experienced by PLWH in Cambodia and the changes in the relationship dynamics between various stakeholders in the HIV care system under the abrupt imposition of COVID-19-related restrictive measures when the pandemic started to spread in Cambodia at the end of 2020. With this understanding, we would propose an intervention that would improve not only the HIV health outcomes, but also the quality of care and engagement of patients through the enhancement of care providers’ relational skills. 

At the heart of this CE work, we embraced the WHO’s conceptual framing of CE, “We live in ongoing systemic processes with one another where every interaction is an intervention and communication is bioactive”, and reckoned that, in order to achieve high-quality and people-centered health services, we should work “with” and not “on” community [7]. This framing, of course, recognized that communication is key to successful intervention, is a powerful tool to shape people’s mindsets and beliefs, and is the foundation for building trust and relationships that will, in turn, affect the care quality [8,9]. This holistic approach to understanding how stakeholders work together in the health system to ensure the successful delivery of health services has also been documented previously [4,10,11,12,13]. 

The need for evidence based on a complex systems model for public health research is not new, but to achieve this researchers and policy makers have to stop trying to solve complex public health problems by assuming a simple, linear causal model; evidence on public health problems focusing on complex systems aiming to change or influence outcomes at a population level should not be based solely on the understanding of a single context within the system [14]. Although the COVID-19 pandemic reportedly caused multiple disruptions in the HIV care services [15,16,17,18], the global response to the COVID-19 pandemic has brought to light the importance of partnership within and between governments and diverse communities (including stakeholders across sectors) to achieve a timely and more effective response [18,19]. 

Conventional responses to a public health crisis, rely heavily on the coordinated actions of technical and clinical specialisms, with the social sciences contributing an understanding of context, culture and the feasibility and acceptability of interventions. This often results in the study of a phenomenon out of its context, which fails to fully capture and integrate subjective experiences in a situation which is dynamic and evolving. In addition, managing uncertainty and the accompanying emotional responses in a crisis cannot be addressed through linear causal mechanisms that are time-bound because the outcome is dependent on the perceptions and actions of all those involved, how they are shaped by the past, and the emergent nature of the event that they are navigating. 

The challenge continues to be understanding how “agency, interconnectedness and unpredictability” influence both evidence generation and evidence translation in systems that are complex for systemic challenges to be addressed by systemic solutions [12]. Therefore, taking the complex systems approach into account, we aim to understand how and to what extent the HIV care services in Cambodia have been affected by the COVID-19 pandemic, and how inter-relationality in the research process could inform the intervention design and ensure their successful delivery.

## 2. Materials and Methods

### 2.1. Study Context

The HIV care delivery model in Cambodia is based on the implementation of the current Community Action Approach (CAA), whose implementation depends heavily on the active participation and involvement of several stakeholders across various levels of the HIV care system in Cambodia [20]. According to the latest estimates, there are over 60,000 PLHIV in Cambodia, among which 30% are registered in Phnom Penh (the capital city) [21]. Here, people can seek HIV care services from one of the many Opportunistic Infections/Antiretroviral Therapy (OI/ART) sites. These sites are under the supervision of the National Center for HIV/AIDS, Dermatology and STD (NCHADS), and there are more than 70 of them across Cambodia. Each OI/ART site transmits their patients’ follow-up or initial visit information for patients who are currently in care or for newly tested positive patients, respectively, to the NCHADS database on a regular basis. At these sites, medical and non-medical staff are responsible for HIV care, support and follow-up. Doctors, nurses and pharmacists are more involved in medically related issues, while counselors, community workers and data clerks work together on socio-emotional support and follow-up of patients. At times, though, some of their roles and responsibilities do overlap. 

### 2.2. Complexity-Informed Iterative Study Processes

#### 2.2.1. Study Design and Sites

We conducted a mixed-method study, with focus group discussion (FGD), in-depth interview (IDI) and quantitative HIV database analyses. In the initial step of the study, we analyzed the data routinely collected at all OI/ART sites in the country using the NCHADS national HIV database system. We examined the number of patients who had lost access to their HIV treatment care, as well as those who were lost-to-follow-up between 2018 and 2020 by OI/ART sites. Based on these numbers, taking into account the relative size (i.e., number of patients in care) of each of the OI/ART site, we purposively selected eight OI/ART sites to be study sites (out of more than 70 sites across Cambodia). Retention and access rates in these eight selected sites appeared to be more severely affected, with high proportions of patients who had dropped out of care and lost access to their HIV treatment over the examined period compared with other OI/ART sites in the country. Among the eight selected sites, half of them (four sites) are in Phnom Penh and the half are in the provinces (one site in Kandal province and another three in Banteay Meanchey province). These eight OI/ART sites accounted for approximately 25% of the total PLWH in Cambodia.

#### 2.2.2. Data Collection Tools and Stakeholders’ Interviews

Prior to the qualitative interviews, the research team reached out to each of the study volunteers (n = 69) individually at selected OI/ART sites, obtained their consent, and completed the Google Form questionnaire on demographic information such as age, sex and educational background. Among the 69 participants there were 43 care providers, 13 patients and 13 high-level stakeholders who were representatives from NCHADS and support organizations. Upon completion of these questionnaires, we arranged for virtual FGD and IDI sessions with each study volunteer according to their preference and schedule. The discussion and interview process followed a similar structure for all stakeholders (care providers, patients and policy-level stakeholders). Once the participants were all present on Zoom, the researchers began by sharing their PowerPoint presentation on linkage, access and retention rates based on the NCHADS database analysis. We showed the overall access and retention rates among patients and then the site-specific access and retention rates from the past four years (2017 to 2020) to the health facility and the policy-level participants. Next, we conducted a root-cause analysis by asking each participant (or group of participants) to identify what were, according to them, the causes behind the access and retention rates which were captured by the database at their OI/ART site). After the root-cause analysis, the study participants were encouraged to share with the research team the solutions that they think would help resolve these issues at their hospital (or clinic), as well as any other measures they may have already taken in order to address these problems. We conducted a total of 18 interviews (15 FGD and 3 IDI). 

All FGD and IDI were conducted using Zoom, and the audio recordings from these sessions were stored (for later transcription) on the University’s online server with restricted access. They were later transferred to the password-protected computer of the designated research team for analysis purposes. The data collection lasted approximately one month, from the end of April to the end of May 2021. About 10 weeks after the intervention implementation in 2021, two brief separate feedback sessions were held for both care providers and PLWH in order to assess benefits and challenges regarding DHI. 

#### 2.2.3. Continued Sensemaking throughout the Development and Implementation of Intervention with Partners and Community Engagement (CE) Experts

Throughout the research process, a team of CE experts, designated Enabling Partners (EP), were commissioned by the WPRO to follow up with and support the research team. The Cambodian research team underwent regular discussion and meeting sessions with the EP (including the CE experts at WPRO) from the initial implementation of our CE project. Besides regular meetings with the EP, each country research team also met one another in Communities of Practice (CoP) meetings where we could hear and learn from other country teams’ experiences. The EP offered their insights and guidance to each country team in their regular, one-on-one scheduled meetings, as well as in the CoP meetings. 

Initial findings from the FGD and IDI indicated that face-to-face communication (medical consultation and counseling sessions) between HIV care providers and PLWH had been greatly interrupted during the pandemic, and that remote counseling and consultation sessions, if those options, existed, would be beneficial for them. These had, of course, been brought up in discussions with the wider expert groups during our meeting sessions before we consensually decided to propose the digital health intervention (DHI) as a means to improve patients’ engagement in care and overall provider–patient interaction during these times of physical and social restrictions. The DHI provided was mainly for communication and capacity building purposes, and consisted of prepaid internet packages and pre-installed social applications (Facebook messenger, Telegram, etc.) and licensed Zoom accounts (one license for each OI/ART intervention site). 

#### 2.2.4. Development of Intervention—Introducing Relational Dimensions within the Digital Health Intervention (DHI)

With support from the EP and CE experts, we incorporated within the DHI several relational skills training sessions, specifically targeting, among others, effective communication and trust building skills for the HIV care providers from the intervention sites. These training sessions were provided to a number of care providers (medical as well as non-medical personnel at the OI/ART sites) at regular intervals over the DHI implementation period. Topics including receptive vs. reactive mode, dealing with one’s emotions, skills in speaking, listening, etc., which were covered and widely discussed among the CE experts and the trainees. There was also practice in applying these skills during the sessions. The relational skills taught were based on the WHO’s guide to building trust and communication [22] and the discussions made with the EP and our local CE experts in conformity with the research objectives (of improving engagement in care, i.e., access and retention rates). It is noteworthy that the DHI provided for study purposes remained at the intervention sites even when the study was concluded. 

Although we previously selected eight OI/ART sites to be study sites where the data collection had taken place, only four out of the eight OI/ART sites was selected to receive the intervention, starting mid-June 2021. However, due to pandemic-related circumstances, one OI/ART site was not able to fully implement the intervention according to plan, so only three OI/ART sites actually received the intervention from mid-June 2021. Later in 2022, the WPRO decided to extend additional support to our CE work and this allowed us to recruit two additional OI/ART sites (out of the original eight study sites) to receive the intervention. Therefore, up until this date, five OI/ART sites (two in the provinces and three in Phnom Penh city) received the intervention in our CE work in total. 

### 2.3. Data Management and Analysis

#### 2.3.1. Data Extraction

The study team obtained permission to use the national HIV database from NCHADS. Each patient in the system had a unique (encrypted) ID code that linked their demographic information to their other HIV-related data files. No identifying information could be found on any data files; only their ID code was used. We followed the instruction to link the patient’s demographic information and their visit data file, which records all the patients’ visits and appointments dates since their HIV diagnosis date. We did this for all eight OI/ART study sites, and restricted the data to a period from 2021 to 2022, when the initial CE work and the extended CE activity started, respectively. 

#### 2.3.2. Analysis

After the transcription process, we took a thematic approach to the qualitative data analysis using NVivo version 1.6.1 (QSR International, Burlington, MA, USA). The scripts of the group and personal interviews were coded into various themes: causes of the problems (changes in access and retention rates in 2020 compared to previous years), including any barriers or challenges described during the pandemic in accessing comprehensive HIV care, solutions used (or endorsed), and other contextual or process knowledge and learning as applicable. The quantitative section of the questionnaire and the NCHADS HIV database were analyzed using STATA 17 (StataCorp LLC, College Station, TX, USA). Means and standard deviation (SD) were calculated for continuous variables, and proportions and percentages for categorical variables. We also calculated the proportions of missed visits by patients and by quarter in 2021 and 2022. 

## 3. Results

### 3.1. Demographics

We interviewed a total of 69 participants: 32 (46%) men and 37 (54%) women. The participants were care providers (n = 43), patients (n = 13) and high-level stakeholders (n = 13). The average age was about 40 years for care provider and patient groups, and much older for the high-level stakeholder group (Table 1).

In terms of education, the majority of care provider and all of the high-level stakeholder participants finished high school or received higher education. However, only 23% (n = 3) of PLWH finished high school or received higher education. More details on the HIV-related factors can be found in Table 1.

### 3.2. Challenges/Barriers to Comprehensive HIV Care during the COVID-19 Pandemic

#### 3.2.1. Among Those Who Accessed HIV Care

Through the stakeholders’ interviews, we were made aware of a multitude of barriers met by our PLWH trying to access HIV care during the pandemic (Figure 1).

Some of the most commonly reported barriers among our PLWH study volunteers included disruptions in travels and fear and anxiety regarding getting sick with COVID-19 (and of exposing HIV status, in certain circumstances). One of our PLWH study volunteers (PG-06) stated that *“…especially since the pandemic, I am so afraid of leaving home. I am already HIV-positive, I fear that I will be the first one to catch this virus…”*. PG-06 also expressed fear and anxiety over not knowing how the HIV infection had been progressing due to suspended blood testing during the pandemic: *“…I want to know my CD4 status but they stop drawing blood (due to Covid-19 and many private laboratories also closed…”*. Another (PG-13) also shared similar concerns, saying *“…I worry too (about not being able to talk with my care providers like before), now all I do is take my medication, I do not know how my disease is going…”*. In addition, PG-08 expressed meeting with financial hardships, making it hard for her to continue coming to receive care at the clinic during the pandemic: *“…I have so many difficulties, alone and jobless and my mother is old…”*. 

#### 3.2.2. Among Those Who Provided HIV Care

There were severe disruptions to HIV services, as well as other pandemic-associated changes that altered the daily operation of the HIV health facilities. Many of the HIV care providers also told the study team about limited counselling/consultation sessions, or even the absence thereof, and the postponement of running blood tests (unless absolutely necessary) at their clinics. These disruptions, of course, could potentially affect patients’ engagement in care and contributed to their fear and anxiety. One care provider (FG-07) told us in the FDG session that *“…we do not meet patients face-to-face now, or sometimes we do but from afar, we do not take vital signs and draw bloods for a while now since the pandemic…, if they (patients) do not have particular health complaint…”*. We had also been informed that the antiretroviral (ARV) regimen, due to a supply halt and increased use of plastic for individual patient ARV medication packaging, was leading to an increase in expenditure for plastic bags for each clinic. FG-13 said that *“…the HIV medication supply had been cut, we are forced to change the patient’s regimen…, shorten the visit intervals…, new regimen or formula are not always well accepted by the patients… and they also create more work for our doctors and counsellors”*. Lastly, challenges in activities to help track patients and bring them back to care had also been exacerbated during the COVID-19 by travel ban, when many community outreach activities had to be suspended. One of our community action workers (FG-44) explained that *“(During the COVID-19 pandemic) I am also not able to go to the community and find the patients who are not keeping their appointments as before (the spread of COVID-19)…”*.

There were also other issues, and although they might not necessarily and directly affect the patients’ care activity, they certainly perturbed the overall functioning of the clinic. For instance, FG-15 also raised the important issue of higher expenses on plastic bags that were used to pack the patient’s medication, in her words *“…the expenditure on PPE had sharply increased (during the pandemic)…, we spent more on plastic bags…, we have to also send the HIV medications to patients who are not able to travel to the hospital to have their medications refilled… I would like to just say that we have to spend so much money”*. Despite the reduced in-person medical visits, heavier workload was another issue commonly raised, not only because many care providers across the hospitals in the country had to take turns becoming COVID-19 frontline workers at various government COVID-19 testing and treatment centers, but also because of the pandemic-related travel bans, as FG-13 mentioned *“…40% of our hospital staffs were in the red zone (no travel allowed), and so the workload had been very heavy for the rest of us…”*. 

### 3.3. Dynamics of the Provider–Patient Relationships during the COVID-19 Pandemic

The relationship between HIV care providers and PLWH in care in Cambodia could be described as strong. Before the pandemic, counseling sessions were opportunities for a private and confidential conversation between PLWH and their care providers. These sessions provided a safe sharing space for many medical and non-medical issues. However, during the pandemic, enclosed indoor space had to be avoided; counseling had to take place in an open space, with limited or zero privacy, or was canceled all together. With fear and anxiety over their own disease status and induced by the pandemic, a constant need to talk about or express these concerns and confirm the correct course of action was overwhelming. Remote counseling and consultation had been raised as one of the most fitting solutions to address this change in the routine of interactions in the pandemic context. According to one of our interviewed PLWH (FB-01), with limited physical meetings with care providers, patients need more assurance from their providers in order to feel they are doing the right thing and staying healthy.

Despite their restrictive interactions, we noted that almost all of our PLWH study volunteers described how they felt that their care providers were always concerned about their well-being and constantly reminded them of their appointments and medication adherence. When asked if they felt any different consulting with their providers remotely (compared to in-person meetings), one of our PLWH study volunteers said *“I do not feel it is different, providers still pay attention the same way (online or in-person meeting), they still ask me if I take my medications regularly, if I had forgotten any. They told me to not forget because they (the medication) are important to my health. I know my providers only want me to be healthy“.* The care providers themselves also revealed that they often talked to their clients as if they were talking to friends and not simply as patients at clinics. According to one care provider (FG-13), *“…If you go through my mobile phone contacts, you see that I saved my patients’ numbers with their names and I talked to and texted them as if they were my friends or relatives, not as my patients…”*. We were also informed of the growing number of PLWH joining the provider-patient support Telegram group chat (part of the DHI) and sharing their thoughts and news with their providers as well as with other PLWH chat members. In addition, the majority of our study volunteers believed that the move from in-person to remote communication and interactions would not have been successful had the prior engagement between patients and care providers not been founded on strong grounds of trust. 

### 3.4. DHI and Relational Skills Training: Benefits and Inconveniences

Toward the fourth and last session of the relational skills training, we asked our care providers (who had been selected to come to join all four sessions of relational skills training) from all five intervention OI/ART sites to complete a short survey on their thoughts on DHI and the relational skills training. The original survey questionnaire can be found in the Appendix A. The results are summarized in Table 2, but overall, we could see that the response to DHI was very encouraging, with many care providers agreeing that using existing social platforms to connect and communicate with PLWH in care had been helpful. It should be noted that three out of the five OI/ART sites were from the previous CE work, but they informed us that their sites were still using the DHI (even after the previous intervention period was over in 2021 and all the project’s prepaid mobile internet plans had been stopped) and that their patient–provider support group chat kept growing. Similarly, the relational skills had been very positively accepted by care providers from all five sites. More details can be found in Table 2. 

The patients who had joined the Telegram chat group with other patients and their care providers, as well as those who had interacted quite often with the care providers using the DHI, also described several benefits of DHI. During the feedback session held about 10 weeks into the intervention for the initial CE work among several PLWH, FB-01 told us that he left voice messages in the group chat or for his care provider when he could not meet to ask those questions in-person during COVID-19, and found that doing so was very convenient: *“…I left messages asking about some medications and he will get back when convenient for him, I found that is highly practical during the pandemic, since I could not go out… And without this option, I think it might be very hard for me”*. Another study volunteer also added that *“Technologies are good, doctor, we can keep in touch especially now that we are still dealing with this COVID-19 pandemic. If we do not need to come to get our medications, we can just consult with our providers using telephone*”. Some believed the communicating with care providers via DHI was a great opportunity for certain patients to open up more to sharing and being informed. Another patient (FB 2) during our feedback interview mentioned that *“I observe that some patients are less shy and more expressive in our group chat, but in person, they are shy and do not want to talk much with us or the providers. Like this, we can obtain more knowledge from what our providers and other patients share in the group chat. That helps keep us informed of matters such as COVID-19 vaccines and other things to keep us healthy”*. 

Although many praised the DHI options, they also raised several points regarding online privacy. Some expressed worries about their voice messages or texts being forwarded to other parties without their consents. In addition, they also noted that the support group chat was sometimes used to share information that was not necessarily related to their health. 

### 3.5. Quantitative Measures of HIV Indicators: Access and Retention

Using the national HIV database from NCHADS, we calculated (at the aggregate level) the percentages of missed visits by patient by quarter in 2022. This was done by dividing the number of missed visits by the total number of appointments given per quarter per each patient. The number will be close to zero if the patient rarely misses their scheduled visit. For noting, a patient who missed their scheduled visit by seven days or more is considered as having lost access to HIV treatment, while those who missed their scheduled visit by more than 28 days will be considered as having been lost-to-care (i.e., they are not retained in care). 

Figure 2 and Figure 3 showed the percentage of missed visits by patient by quarter in 2022 in the intervention and control sites, i.e., each number represents the percentage of missed visits on average by patient per quarter. In Figure 2, for example, in the previous two quarters prior to the intervention, we observed that each patient missed about 3% of their visits by seven days or more in the intervention OI/ART sites, while this number is only around 2% in control OI/ART sites. In the quarter following the intervention, however, the difference in the average number of ≥7-day missed visits per patient between intervention and control sites was smaller (compared to the difference in the previous two quarters). 

Similarly, for retention among PLWH, we saw that in the quarter following the intervention, the average number of ≥29-day missed visits per patient per quarter between intervention and control sites was almost the same, whereas the difference between them was much larger in the previous two quarters (Figure 3). 

## 4. Discussion

The 2021 report by UNAIDS highlighted similar HIV care service disruptions, including the loss of access to HIV treatments and a delay in viral load testing, as some of the COVID-19 impact on HIV [18]. About 38% (n = 5) of the patients we interviewed admitted that it was extremely difficult for them to come to receive their HIV care during the pandemic, especially when there were lockdown measures (second to third quarters of 2021). Similarly, 39% of the care providers (n = 17) we interviewed reported having trouble coming in for work; however, they also acknowledged that they could still come to the hospital or clinic for work despite travel bans if they could provide sufficient documentation for administrative clearance. Disruptions in HIV services, such as delays in viral load testing, were noted by all of our care providers from all eight study sites. A cross-sectional study conducted among tuberculosis (TB) and HIV health professionals in low- and middle-income countries on the impact of COVID-19 on TB and HIV services also reported similar findings to ours in terms of travel and HIV service disruptions, as well as fear of contracting SARS-CoV-2 [23]. Similar to our findings, this survey also reported that 40% and 39% of patients and care providers, respectively, had difficulties coming to health facilities during the pandemic [23].

Building upon these findings, we proposed the DHI and equipped the care providers with relational skill sets that appeared to facilitate provider–patient communications and improve their virtual interaction. The evidence of the benefits of digital solutions in improving treatment adherence and the uptake of health services, including HIV testing services and medication, has also been demonstrated in various studies, as reviewed by Cao et al. [3]. Coping with the HIV pandemic and the COVID-19 pandemic at the same time puts a lot of pressure on people living with HIV, both mentally and physically; their necessary frequent visits to the HIV clinics also make them more susceptible to contracting COVID-19 [18]. Unlike the traditional research method, our research process was not hypothesis-driven and our intervention did not solely focus on the technical aspect of the DHI. Although digital solutions to improve communication and interaction between providers and patients (as part of our intervention) might have been key to keeping patients engaged in care during the pandemic, we also gradually learned from our iterative research processes (through continued meetings with the EP) that fears, anxieties and uncertainties surrounding the pandemic experienced by our healthcare workers and PLWH would also need to be addressed through better communication, or our efforts in improving the latter’s engagement in care would be in ineffective in the long run. In the case study on mental health needs described by Parrish-Sprowl et al. in 2020, the authors described iterative research processes in which various stakeholders were engaged and reengaged to define problems and solutions at their facility; they also noted contexts in which better communication resources could build better physical, mental and social well-being for both the patients and the care providers [24]. In the same report, multiple examples on how effective relational skills were able to empower people and steer tense conversations into productive story sharing were also illustrated [24]. Similar to his work, our take on the DHI relied on the notion that trusting relationships were central to whole health, and hence the DHI would not be complete without communication and trust building skills that would enhance the provider–patient interaction, whether this takes place in-person or virtually. We saw that the majority of the care providers who underwent relational skills training with the CE experts agreed that they learned communication and trust building skills that they had found to be useful in their line of work, and that they were more likely to use these skill sets in the future as well. In addition, Hasson et al. described the mechanism of social “brain-to-brain coupling”, where the perceptual system of one brain is connected to the motor system of another, calling for a shift from single- to multiple-brain reference [25]. This is important to note because this would mean that how individuals communicate and interact influences their actions and thinking, showcasing, once again, the importance of positive communication ecology. Through the relational skills training sessions as part of our intervention in our work, we have contributed to the fostering of positive environments for better communication and positive interactions to drive positive conversation cultures and changes. 

Despite the best efforts made to contribute to robust evidence generation, this study still suffers from several limitations. The findings of our study were limited in terms of their generalizability due to a small sample size in certain subgroups (particularly during the feedback sessions) and the fact that there were only eight OI/ART study sites. However, these study sites were purposely selected based on the HIV database analysis in order to capture those most severely affected by the pandemic. In addition, in the qualitative study, the extent of information obtained from one interview to the next roughly determined whether the sample size was adequate, and from what we gathered the information was repeated after about five to six interviewees in the small subgroup. If we look at our findings pertaining to the negative impact of the COVID-19 pandemic, they were also in line with what had been reported in other studies; we therefore feel confident that our results captured the common negative impact of the pandemic on the HIV care system. We conducted all discussions and interviews via digital platforms, and therefore the traditional method of note taking and observations was extremely limited. Regardless, the interviewees were extremely cooperative and enthusiastic, and our interviewers had also been specifically trained to be prepared for virtual data collection. With assistance from our facility-based workers in arranging the virtual meetings, the FGD and IDI went smoothly, overall. Lastly, our HIV outcome measures were assessed at the aggregate level, assuming, therefore, that everyone from the intervention OI/ART sites actually received the intervention, although this assumption might not be entirely true in reality. However, we also complemented our findings with various observations and qualitative assessments from the relational training and feedback sessions, all of which appeared to show that the DHI and relational skills training really improved communication and patients’ engagement. 

Every day, we are exposed to information or misinformation or engaged in interactions which will, in turn, affect our decisions and health outcomes [9,26]. For this reason, it is important to understand and document relational dimensions in health research. Understanding how people communicate and interconnect helps contribute to the expanding evidence of how the communication ecology and relationships could be integrated into the intervention to produce impactful health outcomes. Our iterative research processes were not driven by a given hypothesis, but rather a progressive quest to learn and find the best-fitting evidence regarding the patterns or contexts which the intervention development and implementation should take into account. Long et al. also argued in favor of this new outlook of health system research, where there is no hypothesis to drive the methods or results [27]. Although on a small scale, our work suggested a synergistic effect of relational skill sets and DHI and how the first could be formulated to strengthen the delivery of the latter. We hope that our work will encourage further health service research where relational aspects and iterative research processes are used to inform intervention development, implementation and evaluation on a larger scale. 

## 5. Conclusions

“Every interaction is an intervention”, and every encounter has the potential to alter our mindsets; every challenge and every thought the community (stakeholders) has and we as researchers have are important and worth documenting as they help to inform the overall process of conducting community engaged research and ensuring the successful delivery and sustainability of the proposed intervention. 

The trusting bonds and relationships fostered between the HIV care providers and PLWH provided a great foundation for testing new means of engagement aiming to improve the healthcare service delivery and patient care experience, pushing the boundaries of understanding and appreciating the non-traditional methods of interaction. This could potentially be useful not only in the context of the current pandemic, but also in the context of introducing the hybrid model of HIV care that might be beneficial even beyond the COVID-19 era and the field of HIV. The pandemic-associated restrictive measures created a research opportunity to better understand the relational aspects of stakeholders within the health system and how these could be used to inform new ways of conducting research and adapting innovative solutions. It is useful to integrate relational skills in nature (for instance listening, communicating or self-understanding skills, etc.) into the intervention, as applicable, to ensure that it will be widely accepted, successfully implemented/delivered and, of course, sustainable. 

Although it has limitations regarding privacy and an obvious need for regulation, we saw how the DHI complemented comprehensive HIV care, reduced in-person meetings during the pandemic, and was widely accepted by both care providers and PLWH for its flexibility and practicability as a tool to help improve the patient care experience and share useful information between patient networks and care providers. Regardless of how care is delivered to patients (in person or remotely), care providers could benefit tremendously from relational skills training, not only because the patients would need constant reassurance amidst times of uncertainty and limited physical interactions, but also so that care providers could have their own space to breathe, reflect, connect and learn new skills regarding trust and relationship building, among others. With these relational skills, we hope to improve the provider–patient communication ecology one interaction at a time. 

## Figures and Tables

**Figure 1 ijerph-20-05247-f001:**
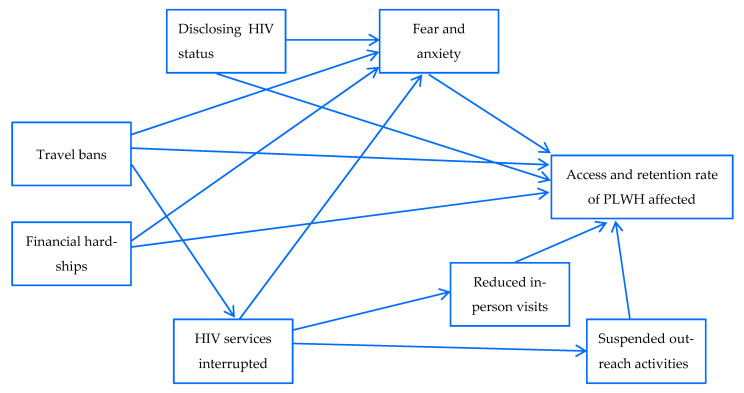
Diagram showing barriers to comprehensive HIV care during the COVID-19 pandemic among the PLWH interviewees, Cambodia CE research project (n = 69), Cambodia, 2021–2022.

**Figure 2 ijerph-20-05247-f002:**
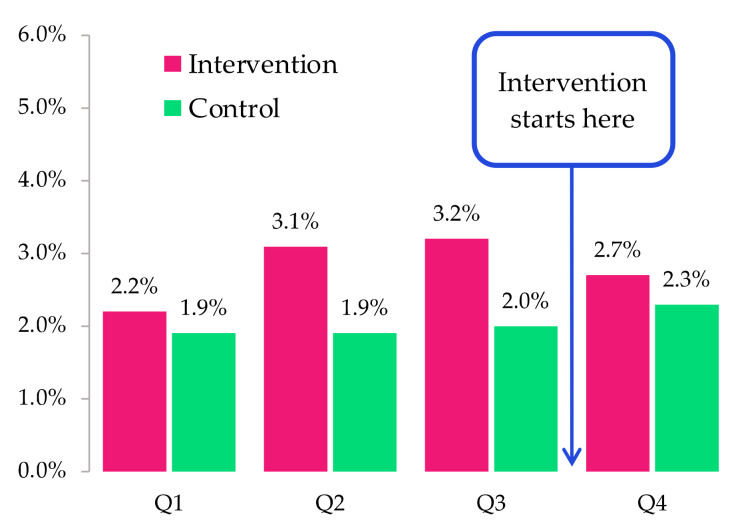
Average number of visits missed by ≥7-day per patient in 2022 by intervention OI/ART and quarters.

**Figure 3 ijerph-20-05247-f003:**
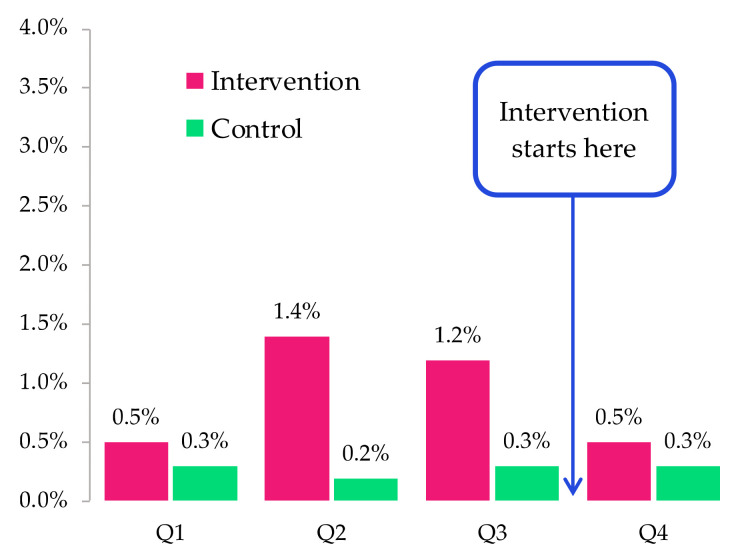
Average number of visits missed by ≥29-day per patient in 2022 by intervention OI/ART and quarters.

**Table 1 ijerph-20-05247-t001:** Demographic characteristics of study volunteers, Cambodia CE research project (n = 69), Cambodia, 2021–2022.

**Characteristics**	**Providers** **(n = 43)**		**Stakeholders** **(n = 13)**		**Patients** **(n = 13)**	
	n	%	n	%	n	%
Gender						
Male	13	30.2	11	84.6	8	61.5
Female	30	69.8	2	15.4	5	38.5
Age (mean, SD)	(41.5, 9.6)		(47.8, 5.3)		(40.8, 13.0)	
Education						
High school or less	6	13.9	0	0.0	10	76.9
Completed high school or higher	37	86.1	13	100.0	3	23.1
Years of working in HIV/AIDS care (mean, SD)	(10.2, 7.6)		(10.8, 7.6)			
Years of living with HIV (mean, SD)					(13.1, 6.3)	

**Table 2 ijerph-20-05247-t002:** Benefits of DHI and relational skills training with care providers from the two additional OI/ART sites, Cambodia CE research project (n = 13), Cambodia, 2021–2022.

Statement	Agree	
	n/N	%
**1. Usefulness of DHI**		
The DHI provided by the project greatly improved our ability to keep in touch with certain patients, even after the COVID-19 situation had improved	10/13	76.9
Many of our patients could benefit from DHI provided to care providers (at intervention sites)	11/13	84.6
Many of our patients have the ability to be connected through DHI provided by the project	8/13	61.5
Many of our patients who have the ability are actually using these technologies (to connect with their care providers)	9/13	69.2
Using DHI to keep in touch with patients should remain as one of the options even after the COVID-19 situation had improved	11/13	84.6
**2. Usefulness of relational community engagement (CE) training**		
Relational CE training sessions within the DHI had been very useful for me	12/13	92.3
Four sessions (of relational skills training) during the intervention period of about two months seemed to be adequate (for me)	11/13	84.6
I believe the relational CE training sessions help me improve my communication and build more trust with my patients	12/13	92.3
I have developed some relational skills I considered useful in order to effectively and compassionately communicate with patients from these training sessions	12/13	92.3
I understood most of the content that had been taught or discussed during the relational CE training sessions	13/13	100
I am confident in using the skills I have learned from these sessions when engaging with patients in the future	12/13	92.3
**3. Perceived need of relational skills**		
I find a lot of uses for these training sessions while engaging with patients through DHI	12/13	92.3
I believe these relational skills remain useful whether the engagement is done in-person or through digital means.	11/13	84.6
I see myself using a lot of the things I learned during these training sessions in the future	12/13	92.3
I am very likely to recommend other providers to undergo this relational skills training in order to improve their communication and trust building skills.	11/13	84.6

## Data Availability

The data presented in this study are available on request from the corresponding author. The data are not publicly available due to ethical reasons and third party involvement. Part of the data was obtained from the National Center for HIV/AIDS, Dermatology and STDs (NCHADS) and are available from the corresponding author with the permission of NCHADS.

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
