# Peer review of "Using Relational Community Engagement within the Digital Health Intervention (DHI) to Improve Access and Retention among People Living with HIV (PLWH): Findings from a Mixed-Method Study in Cambodia"

_ijerph, 2023, doi:10.3390/ijerph20075247_

Round 1
Reviewer 1 Report
This manuscript outlines the use of a relational community engagement within the digital health intervention (DHI) to improve access and retention among PLWH in Cambodia.
Overall the manuscript is clearly written and engaging. It addresses a highly relevant and improtant topic.
Despite the limitations clearly and transparently outlined by the authors, there is a lot of merit to this article.
I would suggest the following major improvements:
1.* The methods and results section includes too much detailed discussion. The methods section includes quite a bit of results and discussion, e.g. lines 170 to 175 in the methods, actually goes into stating some results and discussing them.
*The methods should succinctly state what was done and leave further comment to the proceeding sections. I realize they used a reiterative method to adjust and improve on their methodology but they should outline these in the methods sections without going into the results or dicussion too much. Statement such as ... "Initial findings indicated abc, therefore the survey questions or methods were updated to include xyz..." The key or obvious findings should be stated in the results section but further discussion moved to the discussion section.
2. The discussion could do with more structure and a better progression of arguments.
*I would suggest discussing the results in the same order as they appear in the methods and results section. Mirror the same progression of findings within each section.
I would suggest that the authors:-
a. Start by putting the study into the context of the field and other similar studies and highlighting it's importance as they have already done. I think more references or specific contextualizing and mention of key references is required, e.g. Smith et al. found in their study, using a similar design that.... or the WHO in their policy document... state that...
b. I think the authors should outline all the study limitations within one paragraph or section and mentioned the issue of the small sample size - this is not mentioned. Some of their subgroups came down to less than 10 individuals. Currently the limitations are scattered throughout the discussion and the discussion ends with further discussion of limitations. I think things could be worded more positively e.g. despite the limitation in sample size, our findings were in line with x and y studies from a similar region and with these characteristics that are common to our study and we thus feel confident that......
c. The authors mention the lack of generalizability for example but could discuss, along with this statement, why obtaining such information on a larger scale and gaining a better overall understanding of the issues is important and thus why the study points provides x, y and x positive outcomes.
d. I would end the discussion by emphasizing the merits rather than the shortcomings of the study and the need for more work in the field if this is indeed needed.
3. There are some very relevant and recent references cited but I feel the article could do with more referencing especially to support their arguments about the utility of the study. The authors could do a more critical analysis of the manuscripts in the field and do a better job of placing their study within the context of these other studies and highlighting it's strengths and how it contributes towards any gaps in the current knowledge and points towards the need for more studies to fill these specific gaps...Perhaps, as I have already suggested, the authors could compare other studies or tools that have been initially piloted with small studies but paved the way to more novel methodologies or advantageous outcomes in a similar context or field. I.e., find a parallel scenario that ended up having great utility..
4. The grammar in some of the statements in Table 2 needs to be corrected. Were these the actual statements used in the questionnaire? The first statement in the table, for example, would read better as "The DHI provided by the project greatly improved our ability to keep in touch with certain patients, even after the COVID-19 situation had improved". The full questionnaire, with all the options respondents could choose from, should be provided in a supplementary section.
I recommend the following minor corrections: -
* Line 22 - "...provided the basis..."
*Line 138 - "... they have taken..."
:*Lines 153 & 154 - "...they were not able to obtain counselling from their care providers..."
*Line 418 & 427 - please provide the number of participants along with the percentages as you have done in other sections i.e. 39% (n=?)..
*Line 481 & 482 - omit "supervising", i.e., "...obvious need for regulation supervising"
*Lines 550 & 551 - This reference is in the wrong format. It needs to be reformated to the same style and to include all the relevant similar information as the other citations.
Thank You.
Reviewer 2 Report
Article “Using relational community engagement within the digital health intervention (DHI) to improve access and retention among people living with HIV (PLWH): findings from mixed-method study in Cambodia” aim to understand how and to what extend the HIV care services in Cambodia have been affected by the COVID-19 pandemic, and how the digital health intervention complemented the comprehensive HIV care, improving the healthcare service delivery. The article provides a clear statement of the problem.
There are only a few comments or suggestions:
- What were the reason why the eight selected 01/ART sites have been chosen?
- At sites whom provide healthcare services for PLHIV?
- It’s not clear how many sites were selected to undergo the intervention. Initially were selected eight sites, and later on two additional sites (out to the original eight study sites) were selected
- Line 217 – the number of HIV/AIDS care providers (n=13) don’t match with the number showed in the Table 1 (n=49)
- Line 351 – … about 10 weeks…
- References 8,17 and 21 – Rewrite the titles of the articles (use lowercase letter instead capital letters as initial word letters)
